# Overactivation or Apoptosis: Which Mechanisms Affect Chemotherapy-Induced Ovarian Reserve Depletion?

**DOI:** 10.3390/ijms242216291

**Published:** 2023-11-14

**Authors:** Oren Kashi, Dror Meirow

**Affiliations:** 1The Morris Kahn Fertility Preservation Center, Sheba Medical Center, Ramat Gan 5262000, Israel; oren.kashi@sheba.health.gov.il; 2Faculty of Medicine, Tel Aviv University, Tel Aviv 6997801, Israel

**Keywords:** ovary, primordial follicle, apoptosis, activation, premature ovarian insufficiency, ovarian reserve, chemotherapy

## Abstract

Dormant primordial follicles (PMF), which constitute the ovarian reserve, are recruited continuously into the cohort of growing follicles in the ovary throughout female reproductive life. Gonadotoxic chemotherapy was shown to diminish the ovarian reserve pool, to destroy growing follicle population, and to cause premature ovarian insufficiency (POI). Three primary mechanisms have been proposed to account for this chemotherapy-induced PMF depletion: either indirectly via over-recruitment of PMF, by stromal damage, or through direct toxicity effects on PMF. Preventative pharmacological agents intervening in these ovotoxic mechanisms may be ideal candidates for fertility preservation (FP). This manuscript reviews the mechanisms that disrupt follicle dormancy causing depletion of the ovarian reserve. It describes the most widely studied experimental inhibitors that have been deployed in attempts to counteract these affects and prevent follicle depletion.

## 1. Introduction

Oncology medicine has made great progress in the past forty years, with the ability to diagnose cancer at early stages and increasingly effective individualized cancer treatments. As a result of this success and increased patients’ survival, the long-term effects of cancer treatments have become a major concern, among them ovotoxicity, especially for young cancer patients of reproductive age [1]. Many chemotherapy agents cause a depletion of the ovarian reserve, which may result in POI [2,3]. Active fertility preservation options (oocytes, embryos, and ovarian tissue cryopreservation) are not always suitable for this group of women and have significant limitations [4,5]. However, protection of the ovarian reserve by preventative pharmacological methods that aim to reduce follicle depletion by targeting the mechanisms shown to play a role in chemotherapy-induced PMF loss has great potential for protection of the ovarian reserve. In this review, we present the current knowledge on these mechanisms. We focus on the most researched inhibitors that attenuate components in chemotherapy-induced ovarian reserve depletion mechanisms, their advantages over other inhibitors separately or combined, especially in the comparison between PMF excessive activation and direct PMF oocyte apoptosis pathways. 

## 2. Maintenance of Quiescence and Activation

Dormancy or activation of the PMF are controlled by a complex balance of autocrine, paracrine, and endocrine factors, including growth factors, hormones, transcription factors, and cytokines [3,6,7,8], and are regulated by the interactions between several signaling pathways. Both intrinsic pathways within the oocyte and granulosa cells of the PMF as well as extrinsic factors have been shown to modulate PMF activation (summarized in Figure 1 and Table 1). 

In the PI3K signaling pathway, PI3K and Akt regulate downstream molecules. These regulators link the phosphorylation and activation state of chief modulators’ mammalian target of rapamycin (MTOR) and FOXO3a [28,29] with the growth factor receptor, to initiate the signal transduction cascade [3,30,31]. Studies using genetically modified animal models have demonstrated that the PI3K pathway is a crucial regulator of follicle activation, and the deletion of FOXO3A, MTOR and other proteins such as PTEN and TSC1–2 result in excessive PMF activation, and early follicle depletion [3,32,33,34].

The c-Jun-N-terminal kinase (JNK) pathway, which is a member of the mitogen-activated protein kinase family, was also shown to be involved in PMF activation. The c-Jun protein is regulated at the post-translational level by JNK [19]. This pathway was shown to regulate PMF activation in sheep; pharmacological inhibition of c-Jun phosphorylation using two different inhibitors in bovine ovarian cortex pieces caused a profound inhibition of PMF activation. It has also been speculated that there is crosstalk between the JNK and PI3K-Akt-FOXO3a pathways, in which the JNK pathway activation may induce FOXO3a export from nuclei [19].

Another member of the mitogen-activated protein kinase family–mitogen-activated protein kinases 3 and 1 (MAPK3/1, also called ERK 1/2)–also plays a role in PMF activation, presumably through MTOR signaling. In a three-day mouse neonatal ovarian culture, ovaries that were treated with an ERK1/2 inhibitor had significantly decreased numbers of growing follicles, and also reduced phosphorylation levels of MTOR regulated proteins TSC2, S6K1, and RPS6, indicating inhibition of MTORC1 activity [20].

In addition, extra follicular factors such as the paracrine growth factor Anti Mullerian hormone (AMH), have been shown to negatively regulate PMF activation. AMH-knock-out in mice show increased activation and early loss of PMF reserves [35]. Another series of ex vivo experiments on human and other animal tissue demonstrated increased PMF activation in the absence of AMH, while the addition of the hormone to the culture media blocked this activation [3,23,36,37,38,39].

The Hippo signaling pathway, an extremely conserved pathway [40,41,42] which regulates cell proliferation and apoptosis, is activated when physical disruption of the ovarian cortex occurs [15]. It involves a serine/threonine protein kinase cascade which impedes nuclear access of the key effectors YAP and TAZ by phosphorylation and cytoplasmic retention mechanisms. When ovarian tissue is fragmented, to support better freezing and grafting for fertility preservation in cancer patients, the Hippo signaling pathway is disrupted, leading to the translocation of non-phosphorylated YAP into the nucleus to interact with partners such as TEAD 1–4 transcription factors that target downstream CCN growth factors and BIRC apoptosis inhibitor genes [15,16,17,18,43,44]. 

It has also been demonstrated that physical conditions in the follicles’ surroundings can have an effect on follicle activation. Gavish et al. demonstrated that transplantation of ovarian tissue grafts causes extensive follicle activation and loss of PMF in a multi-species study (bovine, non-human primate, and human), visualized three days post-transplantation [45]. Mechanical stress caused by compression of the oocyte by surrounding granulosa cells with the extracellular matrix (ECM) accompanied with oocytes’ nuclear rotation, was shown to play a role in the maintenance of dormancy in oocytes of PMF in mice [24]. Other studies have suggested that when PMF are clustered together, their activation rate is reduced [25,26] and that loss of several follicles decreased the inhibitory effect on the adjoining PMF in the cluster [27]. 

## 3. Regulation of PMF Survival and Death

In addition to factors that control the balance between follicle activation and suppression are factors which govern survival and death of the PMF. 

The quiescence and survival of PMFs, or their apoptosis, depend on the balance between expression of survival (antiapoptotic) and proapoptotic factors. Members of the p53 family, in particular the p63 isoform TAp63α, were identified as key regulators of PMF oocyte apoptosis following DNA damage [46,47,48,49,50]. The C-Abl protein tyrosine kinase has been shown to act as a ‘switch’ for TAp63 transcriptional activity and the apoptotic pathway in the oocyte following exposure to chemotherapy agent Cisplatin [51]. Tap63 activates apoptosis in the PMF oocyte via B-cell lymphoma 2 (Bcl-2) and Bcl-2-associated X protein (BAX) proteins. These proteins had also been shown to play an important role in the formation of the PMF pool in the prenatal ovary [52,53]. BAX was also shown to be upregulated in PMF oocytes in mice and also in human xenografts following exposure to ovotoxic polycyclic aromatic hydrocarbons, resulting in follicles’ apoptosis [54]. Apoptosis triggering is mediated by TAp73 [55], either directly or through the activation of pro-apoptotic Bcl-2 family members p53 upregulated modulator of apoptosis (PUMA) and phorbol-12-myristate-13-acetate-induced protein 1 (NOXA) [56]. Oocyte-specific deletion of PUMA or of both PUMA and NOXA in mice prevents γ-irradiation-induced apoptosis and can produce healthy offspring, indicating that the protected oocytes are capable of DNA repair and subsequent normal function [56]. 

Other pathways of programmed cell death such as autophagy and necroptosis were shown in recent years to be involved in regulation of follicle survival. Delcour et al. showed that impairment of autophagy, caused by ATG7 and ATG9A variants, which led to a decrease in autophagosome biosynthesis, resulted in a diminished ovarian reserve [57]. Yadav et al. (with autophagy [58]) and Chaudhary et al. (with necroptosis [59]) demonstrated involvement of these nonapoptotic-mediated pathways of programmed cell death in stress conditions within the ovary, hypoxia in particular, which resulted in reproductive impairments including early menopause in mammals.

## 4. Chemotherapy Effects on the Ovary

The impact of chemotherapy on the ovary can be divided into short- and long-term effects. Since most chemotherapeutic agents interfere with the cell cycle of dividing cells, interrupt essential cell processes, and arrest cellular proliferation, their natural targets in the ovary are the proliferating granulosa cells of the growing follicles. As such, they cause the immediate loss of growing follicles at all stages of growth up to preovulatory follicles, resulting in immediate, although sometimes temporary, amenorrhea. The risk of chemotherapy-related POI and long-term amenorrhea depends considerably on the drugs and regimen used. Cyclophosphamide (Cy) for instance is a cytotoxic agent that has been extensively investigated and is widely known to induce amenorrhea [60]. However, the extent of long-term damage to the ovary is contingent with the impact of the drugs on the PMF reserve, and is influenced by several factors, such as the drug family used, dosage, frequency, and route of drug administration and the patient’s age at treatment [61,62]. While total loss of the PMF population can occur, resulting in immediate and permanent sterilization, more common is a partial depletion of the ovarian reserve. In this scenario, the ovary can function post-treatment with a diminished ovarian reserve and fertility can optionally return about six months to a year following completion of treatment [60]. However, this decrease in the PMF pool does shorten the patient’s fertility timeframe, resulting in POI years after treatment [63,64,65,66,67]. The risk of amenorrhea can be predicted by pretreatment AMH levels, with the risk of persistent amenorrhea higher with a lower pretreatment AMH [68,69,70]. Significant depletion in dormant PMF after chemotherapy has been demonstrated in human ovaries [71,72,73,74] as well as in rhesus macaques [75] and mice [14,76,77] (Figure 2). Identifying the mechanisms that play a role in chemotherapy-induced PMF loss has great potential for future protection of the ovarian reserve. There are three major mechanisms that have been proposed to account for PMF depletion, and summarized in Figure 3.
(1)An indirect effect of chemotherapy on PMF via over-recruitment.(2)The indirect effect of chemotherapy on stromal damage.(3)Direct toxicity of the chemotherapy on PMF.

## 5. Mechanisms of Chemotherapy-Induced Follicle loss

### 5.1. The Indirect Effect on Ovarian Reserve: Over Activation of PMFs

Studies indicate that chemotherapy agents such as alkylating agents and platin derivates AA trigger activation and growth of dormant PMFs (Figure 4). This recruitment of PMFs into the pool of actively growing follicles that subsequently undergo atresia or apoptosis is mediated by an upregulation in the PI3K signaling pathway, inducing increased phosphorylation of key proteins AKT, MTOR, and FOXO3A. Overactivation and loss to atresia causes depletion (“burn-out”) of the PMF reserve, which was shown to result in decreased fertility and reproductive outcomes in murine studies [9,11,79,80]. This was further confirmed by studies in which the administration of AMH attenuated ovarian reserve depletion by reducing overactivation of the PMFs, thereby improving fertility in murine models of chemotherapy ovotoxicity [21,22,23] (Figure 5). 

Important results in human ovaries have been presented by Shai et al. They showed follicle activation in vivo in ovaries of patients recently treated (4–12 days) with alkylating agents, as alkylating agent-treated ovaries displayed significant loss of PMFs, and a significant increase in absolute numbers of growing follicles, in addition to activation markers, indicating chemotherapy-induced PMF reserve loss [74]. Further indication was demonstrated in an in vitro study on human ovarian tissue exposed to phosphoramide mustard (PM), a Cy metabolite, which was shown to increase PMF activation and loss [81].

Over-recruitment of PMFs may also explain how protocols involving repeat daily treatments with Cy cause more ovarian damage than single dose treatments, regardless of the total dose administered [82]. Each dose of chemotherapy induces a new burst of follicle activation and growth, and these growing follicles then become atretic or apoptotic as a consequence of subsequent treatment. 

These studies convincingly indicate the pivotal role of PMF activation in chemotherapy-induced ovarian reserve depletion in mammals, and especially in human ovaries.

### 5.2. Ovarian Reserve Depletion Caused by Stromal Damage

Stromal changes such as fibrosis and neovascularization have been suggested as indirect chemotherapy-induced mechanisms of ovarian reserve depletion. Clinical studies have found significant damage to stromal components following exposure to ovotoxic chemotherapy, including obstruction of cortical stromal blood vessels, neovascularization of small, non-mature, disorganized blood vessels in the ovarian cortex surrounding obstructed vessels, and subcapsular focal cortical fibrosis, following exposure to non-sterilizing doses of combined chemotherapy [83,84,85,86]. Increased follicular atresia and collagen deposition [87] and a significant acute reduction of ovarian blood volume and narrowing of the small vessels has been demonstrated in women [88] and in mice after administration of doxorubicin [89]. These results indicate that chemotherapy-induced blood vessel obstruction causes localized ovarian cortical infarcts, resulting in the loss of PMF in that area. Vascular damage is therefore an indirect mechanism by which chemotherapy causes death of the PMFs ([85,90] Figure 3). 

### 5.3. The Direct Effect of Chemotherapy on Ovarian Reserve: Apoptosis

As most chemotherapeutic agents’ mechanisms of action to destroy proliferating cells is through generation of genetic damage, the apoptotic pathway plays a significant role in chemotherapy-induced ovarian injury. 

Double-stranded breaks (DSB) in DNA is one of the main insults induced by cytotoxic agents and the most severe. DSB damage either activates DNA repair pathways that enable cell survival, or when the repair pathways are not sufficiently activated or efficient, apoptotic pathways are triggered leading to cell death [91]. The impact of ovotoxic chemotherapeutic agents on growing follicles has been clearly demonstrated. Almost all classes of chemotherapy induce DNA changes in granulosa cells of growing follicles [51,92,93,94,95], leading either to apoptosis of growing follicles or survival of mutagenic oocytes. Likewise, in vitro exposure of murine ovarian tissue to PM-induced DNA DSBs mainly in oocytes [96]. In most cases, apoptosis of the growing follicles leads to temporary amenorrhea that subsides when new cohorts of growing follicles emerge [90]. When fertilization occurs in oocytes merging from follicles that were recently exposed to chemotherapy, it can result in fertilization failure, high spontaneous abortion rate, and high congenital malformation rate in the embryo—a murine study demonstrated a significant increase in fetal resorptions and more than 10 times the incidence of fetal malformations in conceptions attributable to follicles exposed to Cy during growth [97]. Another study found fetal malformations, chromosomal defects, and also fetal death up to six generations after chemotherapy treatment [98].

Whereas chemotherapy-induced apoptosis in growing follicles is well established, the direct role of apoptosis in dormant PMFs is under debate. According to several studies, chemotherapy induces ovarian reserve depletion directly by causing PMFs to enter into atresia [91,99].

In a human ovarian xenograft model, Cy has been shown to induce DSB in oocytes of PMFs and activate the DNA damage response [100]. These results were confirmed in an in vitro study using ovarian tissue cultured with PM [101,102] and following in vivo Cy injection [103]. Apoptosis of mouse oocytes was similarly seen following both in vivo and in vitro treatment with Cisplatin or Cy [51,101,103], as well as in a xenograft model of human cortex ovaries [104] (Figure 6). 

However, many studies that searched for apoptotic markers in PMF following gonadotoxic chemotherapy exposure did not find any indication for direct cell death [9,10,74,77,79,80] (Figure 6). However, although direct cell death of dormant PMF oocytes is considered questionable and not demonstrated in many studies, many studies still suggest that part of the ovarian reserve depletion is caused by apoptosis. 

### 5.4. Combination of Activation and Apoptosis?

Nowadays, the existing literature consists of conflicting results: many studies have demonstrated chemotherapy-induced PMF activation, while other studies have not found activation. On the other hand, some studies have proposed a direct effect of chemotherapy on the ovarian reserve via PMF apoptosis while many other studies have not demonstrated apoptosis indicators. It is plausible that these conflicting results are the result of differences in the experimental type of system (murine, primates, human), age, timing, type of treatment, as well as different methods of activation or apoptosis detection. 

One study examined DNA damage and apoptosis, as well as activation markers up to 12 h post-Cy exposure [73]. While it is well accepted that this timeframe is sufficient to show apoptosis in oocytes exposed to chemotherapy, it is controversial regarding the evaluation of timing for activation, i.e., 12 h post exposure might be too short. The majority of the studies investigating activation evaluated activation markers 24 h to a few days post-chemotherapy exposure [9,10,11,14,22,23]. 

Another variable that can contribute to inconsistent results is the method used for evaluation and its interpretation. While morphological parameters can be used as an in-dication of follicle atresia [105,106], others measure apoptosis using different markers, such as TUNEL, cPARP, or caspases. While some markers will show apoptotic events, other markers will fail to document apoptosis. 

A common claim raised to negate follicle activation mechanism as a cause of chemo-therapy-induced follicle loss is the steady number of primary follicles present pre- and post-chemotherapy treatment. However, as primary follicles are highly sensitive to chemotherapy, and their number decrease following exposure, the fact that their number remain the same as before the exposure implies follicle activation. 

It seems likely that given the right timing and methods, both mechanisms might contribute to PMF loss. In fact, few publications demonstrated the involvement of both PMF activation and apoptosis in chemotherapy-induced PMF loss. Bellusci et al. showed that Cy treatment triggered both the early markers of DNA damage response as well as the activation of the PI3K follicle activation pathway [107]. Maidarti et al. reviewed the links between the PI3K pathway in PMF and the DNA damage response [108].

In a recent study, Kashi et al. showed a loss of 78% of the PMF reserve 21 days after exposure to Cy. Evaluating both activation and apoptotic mechanisms indicated that most of the PMF depletion was caused by excessive activation, as was shown by increased induction of PI3K pathways and primary follicle proliferation. To a lesser degree apoptosis contributed 20% of the PMF oocytes’ loss 24 h after exposure to Cy [14] (Figure 4 and Figure 6).

## 6. Prevention of Chemotherapy-Induced Ovarian Damage

Gonadotoxic chemotherapies such as alkylating agents and platins are commonly used in the treatment of many types of cancer. In recent years, development of individualized treatments and immunologic and biologic protocols enabled physicians to reduce the use of gonadotoxic protocols. This clinical change may assist in preserving future fertility in cancer patients. When physicians are compelled to use gonadotoxic chemotherapies, several active fertility preservation (FP) techniques have been developed. These include oocyte and embryo freezing prior to chemotherapy administration [109], and ovarian tissue cryopreservation before and even after initiation of chemotherapy protocols [110,111]. Although highly successful, these fertility preservation techniques have significant limitations: these methods are invasive, expensive, not applicable for children (oocyte and embryo freezing), limited in outcomes, and carry a risk of reintroduction of cancer cells (ovarian tissue cryopreservation). Studies have shown that targeting the mechanisms shown to play a role in chemotherapy-induced follicle loss has great potential for protection of the ovarian reserve. Preventative pharmacological methods that aim to reduce follicle loss are non-invasive, available to all ages, and may protect the patient’s natural fertility, making this an ideal potential method for passive FP. Moreover, pharmacological passive fertility preservation can be used side by side with the active methods presented. 

Some of the most investigated agents targeting these mechanisms are reviewed in Table 1. One group of preventative pharmacological methods aims to minimize the depletion of the PMF reservoir, resulting from chemotherapy-induced excessive activation, caused by diminished regulation of PMF dormancy.

The PI3K/AKT/MTOR pathway is a major regulator of PMF activation. As such, the main proteins involved in this cascade are targets for inhibition of such activation, with the purpose of preserving ovarian follicle pool. One of the key proteins in this cascade is the mammalian target of rapamycin (MTOR). Studies that were carried out in mice have shown that inhibition of MTOR by rapamycin or its derivatives (temsirolimus, everolimus) protects the ovarian reserve from Cy- and Cisplatin-induced loss of PMF by blocking the phosphorylation of RPS6 and 4E-BP [10,11,12,13,14,112], resulting in 50% PMF protection as well as improved fertility in mice (Figure 5). In contrast, the deletion of negative regulator proteins PTEN or TSC1 increases MTOR activity in the oocyte, resulting in ovarian reserve depletion, and POI [113,114]. The immunomodulator AS101 was also shown to reduce Cy-induced follicle activation and loss and improve reproductive outcomes via its inhibitory actions on the PI3K pathway, and by reducing apoptosis in granulosa cells of growing follicles [9]. 

Administration of the external follicle suppression factor AMH was also shown to attenuate ovarian reserve depletion up to 50% and improve fertility in murine models of chemotherapy ovotoxicity [14,21,22,23], possibly via renewal of the AMH levels that had dwindled due to reduced secretion following chemotherapy-induced growing follicles loss. The advantage of AMH over other activation inhibitors is that its activity in the adult female is almost entirely restricted to the ovary [115]. Roness et al. also found that the recombinant AMH (rAMH) used in their research did not interfere with the therapeutic actions of chemotherapy treatment [23].

Kashi et al. combined the administration of MTOR inhibitor temsirolimus and rAMH, and found that this combination of inhibitors of follicle activation totally eliminated (100%) Cy-induced follicle depletion [14]. It was speculated by Sonigo et al. that AMH suppression of follicle activation is mediated by a downregulation of the PI3K pathway and inhibition of FOXO3a phosphorylation [22]. However, the fact that both inhibitors had similar degrees of partial protection on PMF numbers, and full protection when combined, suggested that rAMH may act primarily via an alternative PMF suppression pathway to the PI3K pathway, and implies a complementary effect between the two inhibitors. This combination between the two inhibitors needs to be further examined before it can be used in clinical settings. Perhaps another set of experiments in primate or human tissues may confirm the murine results and strengthen this combination as the most efficient way to abolish the gonadotoxic damage.

Another group of preventative pharmacological methods aims to suppress components of the apoptotic pathways in PMF caused by gonadotoxic chemotherapies. The anti-apoptotic membrane sphingolipid Sphingosine 1 Phosphate (S1P) was shown to decrease apoptosis in mouse oocytes of PMFs following in vivo administration of the alkylating agent dacarbazine, thereby protecting fertility [116]. In a human ovarian xenograft model, S1P blocked apoptotic PMF death induced by Cy and doxorubicin and preserved the PMF pool [117]. 

The BCR-ABL tyrosine kinase inhibitor Imatinib was shown to prevent PMF loss caused by Cisplatin. Cisplatin induces DNA damage and apoptosis in PMFs via TAP63 activation, and the administration of Imatinib blocked TAP63 accumulation thereby preventing follicle apoptosis [51]. These results were confirmed in an in vitro culture system and subrenal grafting of mouse ovaries. In this study Kim et al. demonstrated that Imatinib protected oocytes from Cisplatin-induced cell death by inhibiting c-Abl kinase from activating TAp73-BAX-mediated apoptosis [55]. However, Kerr et al. showed that Imatinib did not protect PMF oocytes from Cisplatin-induced apoptosis or prevent loss of fertility in two independent strains of mice [118]. In recent years, a few studies have also challenged this outcome. Salem et al. found that 4–6 weeks of daily injections of Imatinib in mice led to a decrease in PMF counts, smaller number of blastocysts formation with fewer total cells, and a significant shift from inner cell mass to increased trophectoderm cells [119]. It was also shown that the administration of Imatinib to mice significantly reduced the PMF counts and increased protein levels of caspase-3 and α-SMA, suggesting that Imatinib potentially exerts ovarian toxicity via apoptotic processes, similarly to Cy [120]. 

Two more inhibitors of ovarian reserve depletion that do not directly inhibit an activation or apoptotic pathway were recently introduced.; Melatonin, an amine hormone produced by the pineal gland in human and other mammals, is secreted in the ovaries and placenta [121]. This hormone has antioxidant activities of scavenging free radicals and reducing oxidative stress injuries in human oocytes and granulosa cells [122]. Melatonin’s ability to protect the ovarian reserve against ovarian damage caused by gonadotoxic chemotherapies was recently proposed: Liu et al. demonstrated Melatonin inhibits granulosa cell autophagy by activating the PI3K/Akt/MTOR signaling pathway, thereby exerting a protective effect against ovarian damage [123]. Barberino et al. have shown that pretreatment with Melatonin prevents PMF loss and reduces apoptosis and oxidative damage in the mouse ovary following exposure to Cy [124]. In another study, Feng et al. showed that treatment with Melatonin significantly prevented Cy-induced overactivation of PMF by maintaining the plasma level of AMH and subsequently preventing litter size reduction in mice treated with Cy [125]. Recently, Hassanzadeh et al. demonstrated that Melatonin decreases the toxic effects of acetamiprid in mice oocytes, embryos, and ovarian tissue, probably by minimization of its oxidative stress [126]. 

Luteinizing hormone (LH) was also demonstrated to protect the ovarian reserve of prepubertal mouse ovaries from Cisplatin-induced follicular depletion [127]. Rossi et al. showed that in vitro treatment with LH reduced the proapoptotic TAp63 protein levels in PMF oocytes, thus promoting DNA repair in the oocytes. They also showed that when prepubertal female mice were treated with LH and Cisplatin together, ovarian reserve depletion induced by Cisplatin was reduced. In a recent study, Marcozzi et al. characterized proteins involved in oocytes depletion following Cisplatin with or without LH by analyzing culture medium of prepubertal ovarian tissues following treatment. In this study, they confirmed the relevance of LH action on prepubertal ovarian tissues exposed in vitro to Cisplatin and LH and provided novel information about the proteins involved in the apoptotic reaction to Cisplatin [128]. 

The typical characteristics that all these inhibitors have in common is their ability to block major processes in the PMF, either by attenuating oocyte growth or follicle proliferation and subsequent transition to growing follicles, or by preventing cell death pathways. The success in keeping the PMF in their dormant state, thereby preserving the ovarian reserve pool, requires strict quality control measures to ensure undamaged genetic material, such as fertility trials, to test reproductive outcomes including litter size, fetal resorptions, and malformation rate (Figure 5).

It is important to emphasize that almost all the studies using different inhibitor types that were reviewed here were performed in murine systems. Taking a step forward with any of the inhibitors to primates or even human systems may advance this technique to become a candidate for FP. Inhibitors that are already used in clinical settings like rapamycin or temsirolimus have an advantage of safety, as well as rAMH that was shown to not interfere with the chemotoxic activity of Cy in vitro or in vivo [23]. 

To conclude, it seems that all three mechanisms are involved in chemotherapy-induced total loss of PMF. However, after investigating the plethora of research, human studies in particular, but also murine studies, indicate that ovarian reserve loss is mostly caused by excessive activation of PMF [14,23,74,81]. There was also greater protection to the ovarian reserve when inhibitors of activation pathways such as the PI3K pathway and AMH were used. Moreover, when these inhibitors were combined (Kashi 2023), even complete protection was achieved. This set of experiments makes the two inhibitors promising candidates for near future research in humans. Local stromal damage and particularly direct PMF oocyte death have also a significant contribution to the ovarian reserve depletion as was shown by murine studies. However, while inhibitors like MTOR inhibitors and rAMH are considered safer, apoptosis inhibitors may potentially affect chemotherapy treatment, and more research needs to be carried out to reveal the PMF death mechanism.

## Figures and Tables

**Figure 1 ijms-24-16291-f001:**
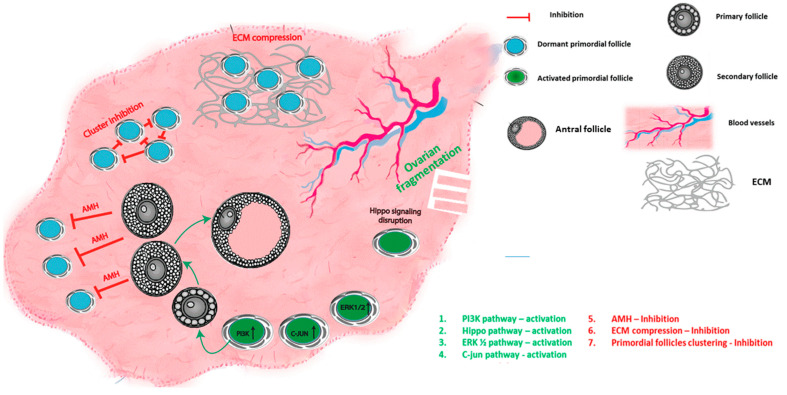
Primordial follicle regulation mechanisms: schematic summary of the different mechanisms that govern primordial follicle activation and quiescence; activation by upregulation of PI3K, Hippo pathway disruption, ERK1/2 and C-jun pathways. Suppression by AMH, compression of the primordial follicles by ECM, and inhibition by clustering of the follicles.

**Figure 2 ijms-24-16291-f002:**
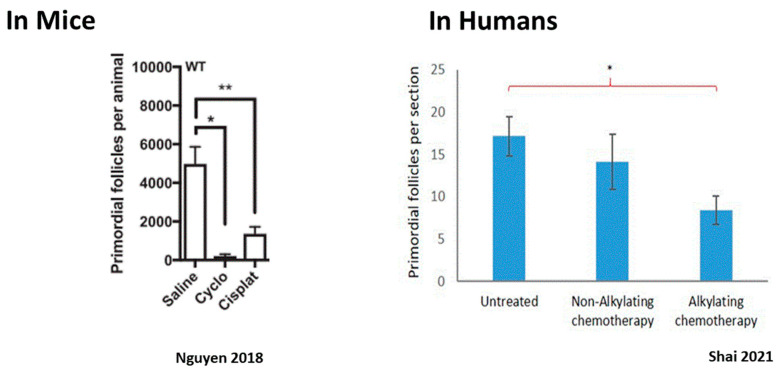
Ovarian reserve depletion caused by alkylating chemotherapy: alkylating chemotherapy induction of PMF depletion in mice and humans. (**Left panel**): PMF depletion following mice ovaries’ exposure to cisplatin and cyclophosphamide. * *p* < 0.05, ** *p* < 0.01. Abbreviations: Cyclo: Cyclophosphamide, Cisplat: Cisplatin. Reproduced from [78]. (**Right panel**): PMF count in ovarian tissue retrieved from women treated with alkylating agent, non-alkylating agent chemotherapy, and untreated women 4–12 days after treatment. * *p* = 0.017. Reproduced with permission from [74]; Elsevier, 2021.

**Figure 3 ijms-24-16291-f003:**
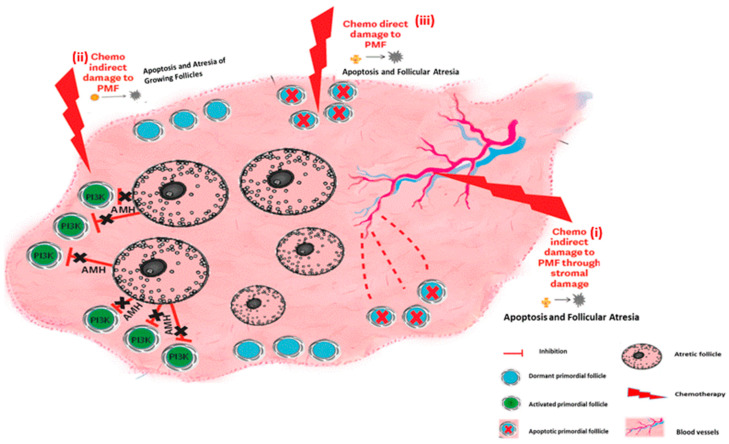
Ovarian reserve depletion mechanisms: chemotherapy effects on the ovary—summary of mechanisms of chemotherapy induced ovarian damage. (**i**) Chemotherapy indirect damage to PMF through stromal damage. (**ii**) Chemotherapy indirect damage to PMF: chemotherapy induces both activation of the PI3K pathway and atresia of growing follicles, thereby abolishing AMH secretion from the atretic follicle, resulting in the removal of the suppression on PMF activation. These two actions cause follicular depletion by massive activation of the primordial follicles. (**iii**) Chemotherapy direct damage to primordial follicles: chemotherapy induces apoptosis directly in the PMF.

**Figure 4 ijms-24-16291-f004:**
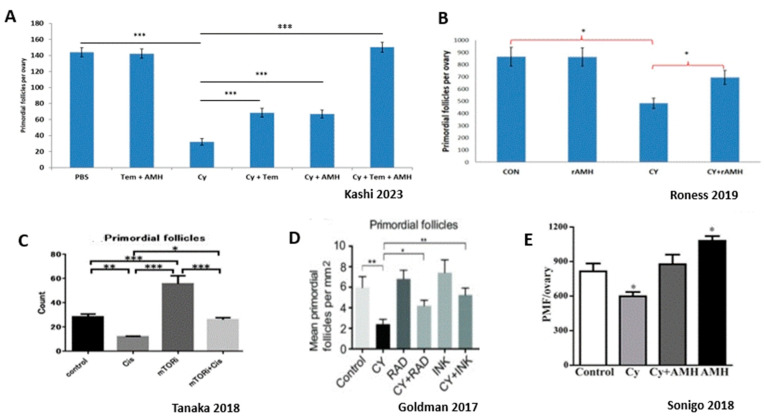
Indirect effect on ovarian reserve: overactivation of primordial follicles. Representative studies demonstrating ovarian reserve depletion caused by excessive primordial follicle activation induced by gonadotoxic chemotherapy. Abbreviations: Tem: Temsirolimus, rAMH: recombinant AMH, Cy: Cyclophosphamide, Cis: Cisplatin, MTORi: mammalian target of rapamycin inhibitor, Rad: MTORC1 inhibitor, INK: MTORC1/2 inhibitor. (**A**) reproduced with permission from [14]; Oxford University Press, 2023. *** *p* < 0.001 (**B**) reproduced with permission from [23]; Springer Nature, 2019. * *p* < 0.05 (**C**) reproduced from [12]. * *p* < 0.05, ** *p* < 0.01, *** *p* < 0.001 (**D**) reproduced from [11]. * *p* < 0.05, ** *p* < 0.005 (**E**) reproduced with permission from [22]; John Wiley and Sons, 2018. * *p* < 0.05.

**Figure 5 ijms-24-16291-f005:**
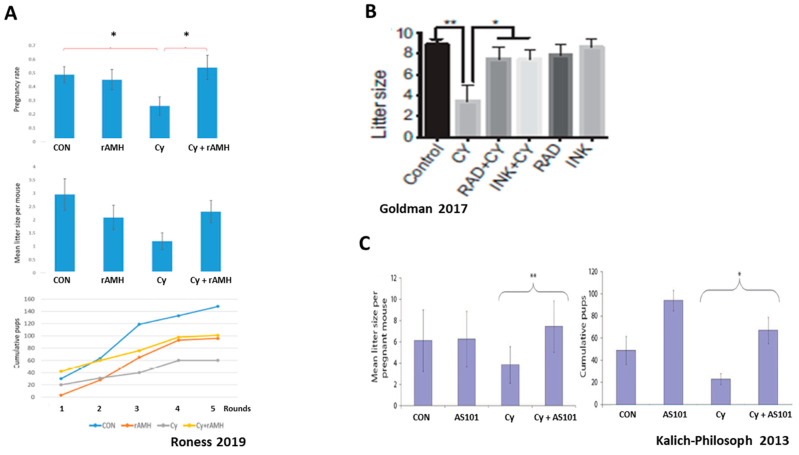
Inhibitors of primordial follicle activation rescue fertility in mice: three representative studies. Abbreviations: rAMH: recombinant AMH, Cy: Cyclophosphamide, Rad: MTORC1 inhibitor, INK: MTORC1/2 inhibitor, AS101: Immunomodulator. (**A**) reproduced with permission from [23]; Springer Nature, 2019. * *p* < 0.05 (**B**) reproduced from [11]. * *p* < 0.05, ** *p* < 0.005 (**C**) reproduced with permission from [9]; The American Association for the Advancement of Science, 2013. * *p* < 0.05, ** *p* < 0.01.

**Figure 6 ijms-24-16291-f006:**
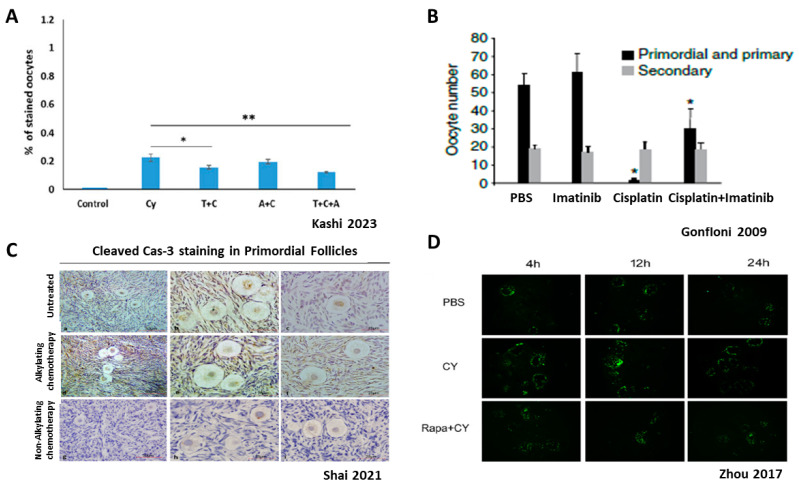
Direct effect of chemotherapy on ovarian reserve: apoptosis. Ovarian reserve depletion caused by direct primordial oocyte death is under debate; A+B demonstrate apoptosis, while C+D show no indication of apoptosis (**A**)—Cleaved PARP staining for apoptoic oocyte, with or without Cy, rAMH or temsirolimus. (**B**)—Imatinib–apoptosis inhibitor, with or without cisplatin. (**C**)—Cleaved caspase staining (apoptosis marker); Representative images of primordial follicles showing no positive staining for cleaved caspase-3 (brown) in oocytes or granulosa cells in sections from untreated women (**a**–**c**), women treated with alkylating agent chemotherapy (**d**–**f**), and women treated with nonalkylating agent chemotherapy (**g**–**i**). Scale bars: (**a**,**d**,**g**) = 50 mm; (**b**,**c**,**e**,**f**,**h**,**i**) = 25 mm. (**D**)—TUNEL staining (apoptosis marker); TUNEL staining conducted in Cy-treated group, PBS group and co-treated group (Cy and Rapa) for apoptosis observation using ovaries removed 4, 12, and 24 h after treatment. Abbreviations: T: Temsirolimus, A: recombinant AMH, Cy: Cyclophosphamide, Rapa: Rapamycin. (**A**) reproduced with permission from [14]; Oxford University Press, 2023. * *p* < 0.05, ** *p* < 0.01. (**B**) reproduced with permission from [51]; Springer Nature, 2009. * *p* < 0.001. (**C**) reproduced with permission from [74]; Elsevier, 2021. (**D**) reproduced from [10].

**Table 1 ijms-24-16291-t001:** Properties of primordial follicles regulation mechanisms.

Mechanism	Major Participants	Mode of Action	Inhibitors	References
PI3K	PI3K, PTEN, Akt, MTOR, FOXO3a	Intracellular signal transduction, activated in response to extracellular signals; promotes proliferation, cell survival, and growth mediated through serine/threonine phosphorylation of downstream proteins	MTOR inhibitors such as Rapamycin, Temsirolimus, Everolimus, INK128, AS101	[9,10,11,12,13,14]
Hippo	YAP, TAZ, TEAD 1–4 CCN, BIRC	Intracellular signal transduction activated when physical disruption of ovarian cortex occurs. Involves a serine/threonine protein kinase cascade which impedes nuclear access of key effectors by phosphorylation and cytoplasmic retention mechanisms		[15,16,17,18]
c-jun	JNK, c-jun	Intracellular signal transduction pathway, regulates proliferation and apoptosis in the oocyte through JNK/c-Jun phosphorylation	JNK inhibitor VII	[19]
Erk 1/2	MAPK3/1, ERK1/2	ERK1/2 signaling is activated in pre-granulosa cells of the primordial follicle, leading to enhanced expression of KITL and then activation of PI3K pathway inside the oocyte	UO126	[20]
AMH	AMH, AMHR, SMAD 1/5/9	Secreted from early growing follicles in the ovary, suppresses primordial follicle activation through signal transduction. Most of the participants apart from SMAD 1/5/9 were not identified yet		[21,22,23]
ECM compression	ECM, pre-granulosa cells, oocytes nuclear rotation	Extracellular matrix together with surrounding pre-granulosa cells apply mechanical stress on PMF oocytes, inducing nuclear rotation, which keeps the PMF in dormant state		[24]
PMF cluster	Primordial follicles	Spatial features of primordial follicles, such as size, pattern of tissue distribution, and clustering, influence the fate of individual PMF to become activated or remain dormant		[25,26,27]

## Data Availability

The data underlying this article are available in the article, and when necessary, via orenkashi1412@gmail.com.

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
