# Peer review of "Overactivation or Apoptosis: Which Mechanisms Affect Chemotherapy-Induced Ovarian Reserve Depletion?"

_ijms, 2023, doi:10.3390/ijms242216291_

Round 1
Reviewer 1 Report
Comments and Suggestions for Authors
This manuscript by Kashi and Meirow aimed to review the known mechanisms of primordial follicle activation and experimental inhibitors of primordial follicle depletion. The review’s stated focus is on the gonadotoxic effects of chemotherapy via over-recruitment of follicles, stromal damage or direct toxicity to primordial follicles. This topic is important for individuals who have a diminished ovarian research due to exposure to chemotherapy. While this is a topic that lends itself to be better synthesized and understood within a cohesive review, this review does not do that, and needs significant revisions in the organization, text and figures to be useful. The focus of the review is unclear, entire sections refer to dated literature, and it reads as a list of points instead of a synthesized overview of the state of the field. Additionally, because the review references only a few original articles from the last few years, it does not add a new or updated perspective to the field. To improve this article, (1) figures must be modified for clarity and cartoons must be redone to make stated points (referring to Fig. 7), (2) primary and contemporary references must be used throughout, especially in tables, (3) since this is not a systematic review, a clear description of why certain pathways, inhibitors and papers were chosen over others is needed, and (4) the authors must provide their unique perspective by synthesizing the literature and providing a comprehensive assessment of the current state of the field including what is needed to advance to the clinic and why these treatments have not yet been implemented. Not all experiments and author interpretations are equal, and the results and conclusions must be weighted in this interpretation. Additionally, there are descriptions of experiments in human and rodent tissue and to rodent treatments. In most cases, the source or host are clearly stated in the experimental descriptions but these are compiled together as if there is no difference in how we should interpret this data.
Listed here are some specific comments to consider:
Use the International Protein Nomenclature Guidelines throughout the text, figures and tables. (https://www.ncbi.nlm.nih.gov/genome/doc/internatprot_nomenguide/)
Many paragraphs are split into 1-3 sentences each. It is unclear if these are separate thoughts from the following sentences. Adjust for proper paragraph structure (topic sentence, development, example(s), summary). This could improve clarity overall.
Fig. 1 is refenced in the abstract. This is unusual and a reference for the figure should be ahead of Fig. 2 in the main text instead of the abstract.
There is minimal utility in Fig. 1, but if it is to be kept, I suggest making the following modifications: There must be full references and permissions for the re-printing of figures in Fig.1. The bars in Fig.1 need a legend to explain the different shading. It is unclear what these bars represent and if it needs another Y axis. The Fig. 1 could be all in color or all in black and white – because the authors are the original contributors of these data, they can adjust the figures from the original paper to be more uniform with each other. Overall, additional details are needed to understand this data without the context of the full manuscript.
The title of Fig. 2 should be changed as it demonstrates mostly inhibitory mechanisms. The activation mechanisms are intrinsic and would be turned on in the absence of the inhibitory cues. It is unclear why the activation mechanisms are represented in this same figure as the inhibitory mechanisms shown. The pathways/mechanisms in Fig. 2 are not demonstrated in a consistent way. The only inhibition with text label in the figure is “cluster inhibition”. All represented examples of inhibitions should be labeled or none of them should be labeled. While the red representing inhibition and green representing activation is appreciated, they are not presented in the same way. AMH is secreted by the growing follicles – should those follicles be red in the same way that the activated follicles are green? The green arrows also seem to be an action from one follicle onto the other (if matching the red inhibitory symbol), but I don’t think that was intended – an explanation of what the arrows represent would be helpful (or remove the arrows). The representation of ECM inhibition is also odd. I challenge the authors to think of how to represent that better.
Table 1: The references should be appropriately listed as if cited in the text. For this to be a useful table, the citations must be of original research, not of reviews.
The appropriate HGNC protein/gene nomenclature for humans or mice must be used in the table. For example, c-jun is “JUN” if referring to the human protein “JUN” if referring to the human gene.
It should be clear what data is used to represent this information, from mouse or human? For example, the shuttling of FOXO3 during primordial follicle activation only occurs in mice.
The Hippo pathway is DISRUPTED in response to alterations in external physical cues, not “activated”. The Grosbois citation is incorrectly used in references the Hippo pathway; there are no data in this paper that demonstrate disruption of this pathway.
The AMH row does not refer to David Pepin’s work demonstrating the inhibitory role in mice. The Durlinger paper was also advanced by additional studies and should be replaced with a contemporary article. Again, these references should be of original research.
“ECM compression” and “PMF cluster” is not a pathway, but is listed as such in Table 1. This is confusing. If the intension of the table is to demonstrate inhibitors of primordial follicle activation, then the table should be re-designed to demonstrate that. Otherwise, the last two rows do not make sense. If these pathways are largely unknown, the first column could be titled “inhibition of primordial follicle activation”, the inhibitors from other rows moved to that column, the “inhibitors” column changed to “pathway”, and in the ECM row, for example, “hypothesized to be Hippo pathway” or “unknown” could be stated.
Melatonin and LH are described in the text as inhibitors but not are missing from the table.
Note that Hippo is a pathway name, not a protein. It should not be capitalized.
The adjective form for rodent, of mouse and of mice, is murine.
PMF is used inconsistently. There are still some sentences that use “primordial follicle” spelled out.
Fig 3. represents action of chemotherapy to induce primordial follicle activation. The title should be changed to represent the point of the figure. It is unclear what the difference between “indirect damage to PMF” and “indirect damage to PMF through stromal damage” is. The stromal damage lightning bolt is on the vessels. The numbers in the figure legend do not refer to anything in the figure. There are images in the figure legend that are not in the figure itself and can be removed, but the primordial follicle with the X on it is not represented in the legend. From the figure image, it appears that chemotherapy directly alters AMH coming from atretic follicles – I do not think this was the intention of the authors. The figure should be modified to represent the intended point.
Fig 4. and Fig. 5., as in Fig. 1 are missing details that define abbreviations, and describe the theme of the figures. Describe why were certain figures were selected for Fig. 4 over Fig. 5. Describe why figures from the same paper were split between the two figures. Fig 6 does not explain the abbreviations or staining (what does the DAB and fluorescence bind to in these figures? What does “% of stained oocytes” mean?). I suggest modifying the figures to use the same consistent abbreviations used in the text.
Some statements over-state the data from the cited articles. For example, lines 212 – 216 describe “chemotherapy-induced blood vessel obstruction causes localized ischemic damage, resulting in the loss of PMF in that area”… The manuscripts cited contain data for pieces of this statement, but no experiments have linked vessel damage as causing PMF loss. It should be clear that this statement is hypothesized by the authors and not yet demonstrated by experimental evidence.
There are references to topics or points that are not explicitly stated in the text. For example in line 261+, the authors mention studies that do not show PMF activation (or depletion?) but it is unclear what this is referring to. The “preventative pharmacological methods” in lines 290+ refer to treatments that are under investigation or proposed but speak as if there are a series of methods that are already used in the clinic. This language must be adjusted to reflect the current state of these interventions which are theoretical (only tested in vivo in rodents) or experimental.
Line 296 states that “preventative pharmacological methods aims to reestablish the primordial follicle reserve”. Reestablishment of a reserve is not possible without a stem cell population and the only way to retain the reserve is by preventing destruction or depletion. I do not think that referring to a stem cell source was the intention of the authors and this sentence must be modified for clarity.
Fig. 7 is challenging to understand. There are different cartoons for primordial and primary follicles within the same figure, and representation of nuclear, cytoplasmic and extrinsic factors within the same plane (or without clear distinction of cellular and organ compartments). There are factors and protein modifications that are not referred to in the text. These cartoons should be original to the review article to synthesize the information from primary literature with a cohesive unique perspective.
Line 316, please clarify what “AMH pool” is referring to.
The final conclusion paragraph is weak. Provide evidence or restate evidence that would indicate that one mechanism of primordial follicle depletion is more pronounced over another. There is no reflection of what additional experiments would support inhibitors to advance to the clinic. What combinations of inhibitors are preferred and why? What has the most promise?
Reviewer 2 Report
Comments and Suggestions for Authors
Article Review: "Overactivation or Apoptosis: Which Mechanism Affects Chemotherapy-Induced Ovarian Reserve Depletion?"
Summary: This literature review delves into the mechanisms responsible for chemotherapy-induced ovarian reserve depletion. The authors explore the various factors implicated in this process and propose potential strategies to safeguard the ovarian reserve from the cytotoxic effects of chemotherapy.
Comments/Revisions: This review offers a comprehensive examination of a highly significant topic, deserving of publication. However, it would benefit from professional editing to rectify a few grammatical errors. For example, using the plural form, "mechanisms," in the title is more appropriate since it encompasses a range of factors contributing to ovarian reserve depletion.
Recommendation: Accept
Comments on the Quality of English Language
Article Review: "Overactivation or Apoptosis: Which Mechanism Affects Chemotherapy-Induced Ovarian Reserve Depletion?"
Summary: This literature review delves into the mechanisms responsible for chemotherapy-induced ovarian reserve depletion. The authors explore the various factors implicated in this process and propose potential strategies to safeguard the ovarian reserve from the cytotoxic effects of chemotherapy.
Comments/Revisions: This review offers a comprehensive examination of a highly significant topic, deserving of publication. However, it would benefit from professional editing to rectify a few grammatical errors. For example, using the plural form, "mechanisms," in the title is more appropriate since it encompasses a range of factors contributing to ovarian reserve depletion.
Recommendation: Accept
Round 2
Reviewer 1 Report
Comments and Suggestions for Authors
The current revision is improved and the authors addressed the specific points that were detailed in the review. However, the authors did not fully consider the main critiques, including
"(3) since this is not a systematic review, a clear description of why certain pathways, inhibitors and papers were chosen over others is needed, "
An introductory paragraph is needed to describe the goal of this review and how certain inhibitor studies were chosen to discuss over others. The main point is not brought up until line 312.
and "(4) the authors must provide their unique perspective by synthesizing the literature and providing a comprehensive assessment of the current state of the field including what is needed to advance to the clinic and why these treatments have not yet been implemented."
More perspective and distillation of results are needed throughout - What makes one inhibitor of a pathway better than another or is a combined treatment most effect? Were there differences in murine versus human ovaries? Were there differences in ages, timing or duration of treatment? What next step experiments would fill this gap in knowledge to get it to the clinic? etc.
The only next set of experiments or studies that needed to be performed before clinical use was stated at the end: 417 "more research need to be carried out to reveal the PMF death mechanism. "
The protein/gene nomenclature did not seem to be used correctly based on the species that was referred to (murine gene names are lower case and mTOR = MTOR (human protein, italicized for human gene)). Please review these.
Fix this sentence 411:
Moreover, when these inhibitors were combined (Kashi 2023), even complete protection was achieved, . makes the two inhibitors promising candidates for near future research in human.
